# Different Sources of Mesenchymal Stem Cells for Tissue Regeneration: A Guide to Identifying the Most Favorable One in Orthopedics and Dentistry Applications

**DOI:** 10.3390/ijms23116356

**Published:** 2022-06-06

**Authors:** Victor J. Costela-Ruiz, Lucía Melguizo-Rodríguez, Chiara Bellotti, Rebeca Illescas-Montes, Deborah Stanco, Carla Renata Arciola, Enrico Lucarelli

**Affiliations:** 1Biomedical Group (BIO277), Department of Nursing, Faculty of Health Sciences, University of Granada, Avda. Ilustración 60, 18016 Granada, Spain; vircoss@ugr.es (V.J.C.-R.); luciamr@ugr.es (L.M.-R.); 2Instituto Investigación Biosanitaria, ibs.Granada, C/Doctor Azpitarte 4, 4^a^ Planta, 18012 Granada, Spain; 3Regenerative Therapies in Oncology of the Osteoncology, Bone and Soft Tissue Sarcomas and Innovative Therapies Unit, IRCCS Istituto Ortopedico Rizzoli, 40136 Bologna, Italy; chiara.belloti@ior.it (C.B.); enrico.lucarelli@ior.it (E.L.); 4Orofacial Development and Regeneration, Institute of Oral Biology, University of Zurich, 8032 Zurich, Switzerland; deborah.stanco@zzm.uzh.ch; 5Laboratorio di Patologia delle Infezioni Associate all’Impianto, IRCCS Istituto Ortopedico Rizzoli, Via di Barbiano 1/10, 40136 Bologna, Italy; 6Laboratorio di Immunoreumatologia e Rigenerazione Tissutale, IRCCS Istituto Ortopedico Rizzoli, Via di Barbiano 1/10, 40136 Bologna, Italy; 7Department of Experimental, Diagnostic and Specialty Medicine, University of Bologna, Via San Giacomo 14, 40136 Bologna, Italy

**Keywords:** tissue engineering, bone, regenerative medicine, mesenchymal stem cells, adipose tissue, bone marrow, dental tissue, orthopedics

## Abstract

The success of regenerative medicine in various clinical applications depends on the appropriate selection of the source of mesenchymal stem cells (MSCs). Indeed, the source conditions, the quality and quantity of MSCs, have an influence on the growth factors, cytokines, extracellular vesicles, and secrete bioactive factors of the regenerative milieu, thus influencing the clinical result. Thus, optimal source selection should harmonize this complex setting and ensure a well-personalized and effective treatment. Mesenchymal stem cells (MSCs) can be obtained from several sources, including bone marrow and adipose tissue, already used in orthopedic regenerative applications. In this sense, for bone, dental, and oral injuries, MSCs could provide an innovative and effective therapy. The present review aims to compare the properties (proliferation, migration, clonogenicity, angiogenic capacity, differentiation potential, and secretome) of MSCs derived from bone marrow, adipose tissue, and dental tissue to enable clinicians to select the best source of MSCs for their clinical application in bone and oral tissue regeneration to delineate new translational perspectives. A review of the literature was conducted using the search engines Web of Science, Pubmed, Scopus, and Google Scholar. An analysis of different publications showed that all sources compared (bone marrow mesenchymal stem cells (BM-MSCs), adipose tissue mesenchymal stem cells (AT-MSCs), and dental tissue mesenchymal stem cells (DT-MSCs)) are good options to promote proper migration and angiogenesis, and they turn out to be useful for gingival, dental pulp, bone, and periodontal regeneration. In particular, DT-MSCs have better proliferation rates and AT and G-MSC sources showed higher clonogenicity. MSCs from bone marrow, widely used in orthopedic regenerative medicine, are preferable for their differentiation ability. Considering all the properties among sources, BM-MSCs, AT-MSCs, and DT-MSCs present as potential candidates for oral and dental regeneration.

## 1. Introduction

The concept of stem cells is over 150 years old. In 1868, Gourjon et al. first discovered the presence of a stem cell population in the bone marrow able to de novo generate ectopic bone in heterotopic sites on rodents [1]. Similarly, Friedenstein et al. [2,3,4], demonstrated the ability of a subpopulation of bone marrow non-hematopoietic cells to fast adhere to plastic and to form colonies (CFU-F) that can differentiate into various mesenchymal lineages, such as adipocytes, chondrocytes, and osteoblasts. In the 1990s, the stem-like population in marrow aspirate became well known by the works of Pittenger et al., which led to them being named as mesenchymal stem cells (MSCs) by Caplan [5,6]. Therefore, the acronym MSC was used to represent “mesenchymal stem cells”, “mesenchymal stromal cells”, and “multipotent stromal cells”. In fact, because of their heterogeneous nature, consisting of a mix of differentiated cells and a small portion of progenitors cells, and the lack of in vivo data to demonstrate self-renewal capacity, the nomenclature used to define MSCs is still debated today and continuously updated from most leaders in the field [7,8,9,10]. In 2006, the Mesenchymal and Tissue Stem Cell Committee of the International Society for Cellular Therapy (ISCT) offered a position statement to clarify the terminology: “multipotent mesenchymal stromal cells”. They defined the minimal criteria to identify MSCs as plastic adherence cells with clonogenic ability, expression of cell surface markers (CD73, CD90, CD105), and lack of hematopoietic and endothelial markers (CD45, CD34, CD14, CD11b, CD79α, CD19, HLA-DR), and in vitro tri-lineage differentiation potential into the mesodermal cell lineages [11].

MSCs have been identified and isolated from almost all tissues, including bone marrow, adipose tissue, teeth, amniotic fluid, umbilical cord, liver, tendon, and heart [12,13,14,15,16,17]. However, being MSC-derived from a tissue-specific stromal vascular fraction, they differ in terms of global gene expression, phenotype, proliferation rate, and differentiation ability [18,19,20,21,22,23]. Moreover, the discovery of the ability of MSCs to trans-differentiate into ectodermal and endodermal cell lineages and the inclusion of novel surface markers (CD165, CD276, and CD82) clearly demonstrate that their biology is still not completely understood [24]. In the current era of cell therapy and cell transplantation, the most useful MSC characteristic reported is the non-immunogen profile and the ability to regulate the surrounding immune microenvironment via cell–cell interaction and paracrine activity [25]. Due to the low expression of the Major Histocompatibility Complex (MHC) I and lack of the MHC II, MSCs have strong immunosuppressive effects against T-cell alloreactivity and proliferation [26]. Additionally, MSCs exert an immunomodulatory effect on dendritic cells, natural killer cells, monocytes, macrophages, and B lymphocytes [27,28]. It has recently been proposed that a proinflammatory environment mediated by cytokines such as tumor necrosis factor-alpha (TNF-α), interferon-gamma (IFN-γ), interleukin (IL)-1, and IL-17 provoke the MSC transition from an activated state to a resting state. Activated MSCs can communicate with the surrounding immune cells by secreting molecules, such as prostaglandin E2 (PGE2), transforming growth factor-beta (TGF-β), tumor necrosis factor-alpha (TNF-α), interferon-gamma (IFN-γ), and IL-6. Other major actors are represented by MSC membrane molecules such as programmed cell death ligand 1 (PD-L1), human leukocyte antigen-G1 (HLA-G1), CD40, Jagged-1, intercellular adhesion molecule 1 (CD54/ICAM-1), and vascular cell adhesion molecule 1 (VCAM-1). Interestingly, the release of exosomes and macrovesicles by both resting or activated MSCs can be involved in the propagation of the signal by body fluids to distant sites [28].

All these properties enable MSCs to guarantee tissue homeostasis and regeneration and are responsible for their therapeutic efficacy, making them an ideal source for cell-based therapies, including allogenic transplantation [29]. Among the various stem/stromal cell types, MSCs obtained from bone marrow (BM-MSCs) and adipose tissue (AT-MSCs) remain the most widely used for regenerative purposes [30]. More recently, thanks to the easy and relatively large accessibility of dental-derived stem cells (DSCs), dental pulp stem cells (DP-MSCs), MSC from the pulp of exfoliated deciduous teeth (SHEDs), apical papilla (AP-MSCs), periodontal ligament (PL-MSCs), dental follicle precursor (DFP-MSCs), and Gingival Mesenchymal Stem Cells (G-MSCs) are emerging as an attractive cell sources for bone and dental tissue regeneration [16,31,32,33]. Although they are all MSCs that meet the ISCT minimal criteria, a series of comparative studies have demonstrated differences among AT-MSCs, BM-MSCs, and DSCs. In the absence of a complete understanding of the MSC phenotype and their biological function and characterization, including their tissue-specific niche environment, several issues still remain to be addressed to free clinical translation of cell therapies. It should be noted that the surface markers used to characterize the MSC are not “stem” specific but rather confirm the fibroblast profile [9]. For this reason, along the search for a safe and easily accessible MSC source, many studies are currently focused on exploring different MSC-marker profiles and regenerative capability. More recently, the advent of single cell “omics” technologies is allowing more accurate MSC population identification in complex tissues [34,35,36,37,38].

## 2. Mesenchymal Stem Cell Sources

### 2.1. Bone Marrow Stromal Cells (BMSCs)

Bone marrow MSCs are heterogenous cell populations located in the medullary stroma of bone marrow. They are traditionally collected through gradient cell separation or directly for plastic adherence in vitro (Figure 1). Other isolation methods include flow cytometric methods such as fluorescent- or magnetic-cell sorting [39,40,41]. Although the harvesting procedure is simple and convenient, it causes patient discomfort because it is highly invasive, painful, and requires general anesthesia administration and hospitalization. Human BM aspirates contain only 0.001–0.01% of total stromal cell population, and their characteristics are highly dependent on the age, sex, and the pathological conditions of the donor [5,42,43].

In general, BM-MSCs show the typical immunophenotype profile defined by the ISCT criteria (Table 1). In addition, the expression of the stromal antigen 1 (STRO-1) has been defined as a specific marker for BMSCs at early passages in culture. Other studies have characterized distinct subpopulations with higher MSC potential. For example, Samsoraj et al. [44] identified CD49a, PDGFRα/β, EGF receptor, IGFR, and STRO-3 as a panel of markers to be included in the isolation criteria for multipotent BM-MSCs. In particular, the positive expression of CD271/NGFR, CD140a/PDGFRα, and Sca-1 has been associated with higher bone regenerative potential. In addition, evidence for a non-homogenous MSC profile in the bone marrow, but rather several subpopulations, was recently demonstrated by using rigorous single-cells analyses from different stromal gene-report mice (LEPR^+^, CD51^−^/Sca^+^, PDGFRα^+^, Col2^+^) in which osteogenic and adipogenic MSC subsets were identified in the BM niche [37,45,46].

Furthermore, their activity relating to immune modulation is another crucial aspect useful in cell-based therapy to reduce inflammation [47,48,49,50].

Recently, a new formulation based on BMA clot (bone marrow aspirate) has also been described as a source of MSCs. In this regard, a study has observed that BMA clot from both young and older donors is a potent source of MSCs that serves as a biological scaffold with regenerative properties [51].

**Table 1 ijms-23-06356-t001:** MSC markers reported in literature.

MSC Type	Markers	References
BM-MSCs	CD73, THY1/CD90, CD105, CD146/MCAM, CD29, CD44, STRO-1, OCT4, NANOG	Dominici et al., Samsonraj et al. [11,43].
AT-MSCs	DPP4/CD26, PDGFRa, CD29, CD34, SCA1, CD55, THY1/C90, CD24, BMP7, PI16, WNT2, ANXA3	Merrick et al. [37].
DPSCs	CD44, CD90, CD105, CD73, STRO-1	Mattei et al. [52]

### 2.2. Adipose Tissue Derived Stem Cells (AT-MSCs)

AT-MSCs are embedded in the extracellular matrix between the adipocytes. Obtaining these cells for culture and amplification begins with a lipoaspirate. The lipoaspirate is washed to remove blood and fat residues, then processed with collagenase type I to release the cells from the connective tissue. The product of the enzymatic digestion is centrifuged to separate the lipidic portion from the stromal component, usually regarded as the Stromal Vascular Fraction (SFV). Subsequently, the erythrocytes, the remnants of cell digestion and cell clusters, are eliminated and the MSCs may be separated from the endothelial cells and leukocytes through plastic adherence in selected medium (as for BM-MSCs) or by magnetic cell sorting mechanisms. Finally, to guarantee the correct identification of the mesenchymal cells, tests based on either immunofluorescence or flow cytometry are carried out [53] (Figure 1).

### 2.3. Dental Tissue Derived Stem Cells (DT-MSCs)

A clonogenic population of dental mesenchymal stem cells was first isolated from the pulp (DP-MSCs) of human third molars by Gronthos et al. [16] in 2000, and since then MSCs have also been isolated from other dental tissues.

DT-MSCs comprise all the MSCs that can be recovered from different sites in the oral apparatus. There are eight known major populations of dental tissue-derived MSC: DP-MSCs, SHEDs, PL-MSCs, DFP-MSCs, alveolar bone-derived mesenchymal stem cells (AB-MSCs), stem cells from apical papilla (AP-MSCs), tooth germ progenitor cells (TP-MSCs), and gingival mesenchymal stem cells (G-MSCs) [54]. In the following paragraphs, a brief description of the MSCs harvest site, along with methods of isolation and insights into the most relevant findings regarding each MSC population, will be given (Figure 1).

#### 2.3.1. Dental Pulp Stem Cells (DP-MSCs)

Dental pulp (DP) is a vascularized and innervated connective tissue of ecto-mesenchymal origin developing from dental papilla [55]. It is placed in the central pulp cavity and limited by the dentin of each tooth. DP performs several functions, such as protective, formative, nutritive, sensory, and maintenance of teeth viability [56]. Therefore, it is mainly composed of MSCs, neural fibers, blood, and lymphatic vessels [57].

DP-MSCs were the first human dental stem cells isolated and characterized [16]. They have been widely studied due to easy accessibility, low invasive harvesting, and great proliferative and multilineage differentiation potential [58]. DP-MSCs can be generally isolated from the pulp of the third molar. First, the crown is cut open and the pulp is extracted from the tooth chamber. Then, cells are harvested by mechanical or enzymatic digestion. The mechanical methodology consists of culturing cells directly from small dissected fragments of the dental pulp, whereas the enzymatic technique is generally based on the use of collagenase type I associated with dispase to dissociate single cells from the tissue [59]. DP-MSCs can be characterized through immunophenotype using flow cytometric analysis of the same surface markers that are used for BM-MSCs and AT-MSCs [60].

Due to the capability to differentiate between several types of cells, in vivo and in vitro, DP-MSCs are widely used in regenerative medicine, mainly in preclinical trials and research, sometimes associated with biomaterials for tissue regeneration. Thus, they are useful in tissue repair and engineering, mostly in dental pulp regeneration, tooth reconstruction, periodontal regeneration, bone tissue restoration, skin wounds and burns, angiogenesis and vasculogenesis, neuronal and skeletal tissue damage repair, and endocrinology. DP-MSCs also have a great influence on angiogenesis, as recently reviewed Mattei et al. [61]. It is also known that DP-MSCs have immunomodulatory properties that provide beneficial effects in immune disorders [62].

#### 2.3.2. Stem Cells from Human Exfoliated Deciduous Teeth (SHEDs)

DT-MSCs can be isolated from the pulp tissue of exfoliated deciduous teeth. Miura et al. [63] were the first to successfully isolate MSCs from the DP of SHEDs. Because they are an immature cell population, they also go by the name of immature DP-MSCs. SHED morphology is similar to fibroblast-like cells. SHEDs isolation is the same as for DP-MSCs. These stem cells can be used in regenerative medicine for dental, neuron, and skin tissue engineering [64,65], alone or in association with scaffolds [66]. They also show immunomodulatory properties; thus, they may be beneficial for treating immune disorders [67].

#### 2.3.3. Periodontal Ligament Stem Cells (PL-MSCs)

The periodontal ligament is a soft connective tissue between the cementum and the alveolar bone. It helps to maintain and restrict teeth within the jaw. It plays an important role in supporting the teeth as well as in the nutrition, homeostasis, and repair of damaged tissue [68]. This multipotent cell lineage was first isolated by Seo et al. [68]. Their isolation was performed by enzymatic digestion, as in the procedure for obtaining DP-MSCs and SHEDs. PL-MSCs are widely utilized for periodontal and cementum regeneration [69]. These cells have also been considered for new bone mineralization or immunomodulation treatments; future applications are currently under study [70].

#### 2.3.4. Dental Follicle Progenitor Cells (DFP-MSCs)

The dental follicle is a tissue of ecto-mesenchymal origin that surrounds the tooth germ as a protective sac. Its main function is to regulate the eruption of the developing tooth; additionally, it is the progenitor of the periodontium [71]. DFP-MSCs are obtained from the attached follicle of the tooth at the beginning of the root formation, which was previously assessed radiographically through enzymatic digestion. These cultures and cells can be characterized by flow cytometry. In spite of the fact that DFP-MSCs have the potential to differentiate into various types of cell lineages, their clinical application nowadays is limited to dental tissue engineering [72].

#### 2.3.5. Alveolar Bone-Derived Mesenchymal Stem Cells (AB-MSCs)

The alveolar bone is attached to thickened ridge tissue in the basal bone of the jaw, and it is embryonically derived from the dental follicle. It is perforated by many foramina, which transmit nerves and vessels. Matsubara et al. [73] were the first to successfully isolate AB-MSCs, as reported in 2005. The method of extraction is similar to that of other dental tissues [74]. A potential therapeutic use of AB-MSCs in tooth engineering has been described, although further studies are needed to elucidate clinical applications [75].

#### 2.3.6. Stem Cells from Apical Papilla (AP-MSCs)

The apical papilla is the soft tissue at the apices of developing permanent teeth. It assists in tooth formation and becomes the dental pulp tissue. Thus, AP-MSCs can only be isolated at certain stages of tooth development [75]. Sonoyama et al. [76] were the first to report the isolation of such cells. Two reliable methods to obtain them exist. The first, consists of the tissue with collagenase type I followed by culture of the obtained cells in culture dishes. The second is based on the culture of small pieces of apical papilla samples on culture dishes, without digestion [77]. AP-MSCs are characterized by flow cytometry to identify specific stem cell surface markers. Therapeutic implications of AP-MSCs are those related to the repair and the regeneration of different tissues such as pulp, dentin, root, periodontal tissue, bone, neurons, and blood vessels. Their use in immunotherapy has recently been described [77,78].

#### 2.3.7. Tooth Germ Progenitor Cells (TP-MSCs)

In humans, organogenesis of tooth germs begins at 6 years old. TP-MSCs are a novel multipotent cell population isolated from the tooth germ of an impacted third molar. This population was first described by Ikeda et al. in 2008 [79]. TP-MSCs are usually extracted due to orthodontic problems [80]. The cell isolation is the same as described in other dental tissues. They showed similar potential lineages to the rest of the dental tissue stem cells [75]. Therefore, TP-MSCs can be used in dental engineering, becoming an alternative source of adult stem cells. Further research is needed to fully understand their immunomodulatory properties in order to test if they can be used as a source of heterologous cells [81].

#### 2.3.8. Gingival Mesenchymal Stem Cells (G-MSCs)

Histologically, the gingiva is a squamous epithelium underlined by a lamina of connective tissue that binds to the bone. It is also covered by a keratinized epithelium to guarantee a normal oral function. Thus, this tissue is composed of a wide range of cell types and extracellular matrix components [82]. Anatomically, it is located in the periodontium, surrounding the edge of the teeth and enclosing the tooth in the alveolar bone. Gingiva is the unique tissue extended in the alveolar ridges, recognized as a biological mucosal barrier. Hence, it makes an easily and accessible tissue to isolate G-MSCs with minimal disturbance during a regular oral surgical procedure [54]. G-MSCs are obtained from a connective tissue biopsy extracted surgically. Single cells are obtained by enzymatic digestion, and characterization is carried out to prove G-MSCs identity [19,83]. G-MSCs provide beneficial factors in different tissue repair and regeneration types, such as skin wound healing, tendon injuries, periodontal damage, bone defects, peri-implantitis, and oral mucositis. In addition, an antitumor effect has been reported and, because of their immunomodulatory properties, G-MSCs are used therapeutically in autoimmune diseases such as rheumatoid arthritis, among other clinical indications [19,83,84].

## 3. Source Comparison

Table 2 shows an overview of the properties of MSCs obtained from bone marrow, adipose tissue, and dental tissue.

### 3.1. Proliferation

Proliferation is one of the main characteristics to be considered for clinical application of ex vivo-expanded MSCs. In 2009, Riekstina et al. [85] proved that both human BM and AT-MSCs were able to double in a period of 24–48 h. Similar results were published by Heo et al. [86] and Zong et al. [87] regarding BM and AT-MSCs. Danisovic et al. [95] pointed out in an assay with human AT-MSCs that the proliferation index increased until passage 10, then showed a slight decrease. A high proliferative state was maintained until passage 25, while there was a reduction of 32% in proliferation at passage 30. A higher proliferation rate was shown in MSCs from lipoaspirates by Hakki et al. [96], in a comparison carried out with human palatal adipose tissue (PAT-MSCs), but with no statistical differences. In a study in which MSCs from seven different tissues, including those from dental tissue (DP-MSCs, PL-MSCs, G-MSCs, DFP-MSCs) and two sources from somatic tissue (BM-MSCs and AT-MSCs) were tested, cells from dental tissue MSCs proliferated faster than BM-MSCs and AT-MSCs [92]. In fact, Stanko et al. [97] established that DP-MSCs are cells with a high proliferative potential. Moreover, TP-MSCs cells have remarkable proliferation in vitro and can be expanded and maintained for almost 60 doubling populations with a high rate of proliferation [69]. In comparing sources only from dental origin, higher proliferation rates were also found by Angelopoulos et al. [98] in G-MSCs when compared with DP-MSCs. Thus, AT-MSCs and BM-MSCs are both good sources for dental applications of MSCs to have a good expansion of the cells, although it is preferable to use DT-MSCs because they have better proliferation rates than the other two sources. In summary: among AT-MSCs, lipoaspirate is a better source than adipose tissue from the palate. In the DT group, G-MSCs are superior to DP-MSCs whenever good proliferation results are sought.

### 3.2. Migration Capacity

Another important feature of MSCs is their migration. For example, in gingival regeneration, the migration potential of cells is important in order to achieve gingival augmentation and healing [100,101]. The same quality is required in dental pulp regeneration [98]. Some authors, such as Jeon et al. [102], have performed migration assays to compare this capacity between different sources. Thus, they determined that BM-MSCs and AT-MSCs have similar migration capacities. This ability is known to be higher in G-MSCs in comparison with other DP sources [98]. Because both BM and AT-MSCs have good migration capacity, the clinician should use the one that is more convenient to obtain. Concerning the dental sources of MSCs, it could be a better option to use G-MSCs instead of DP because they have better migration quality.

### 3.3. Clonogenicity

Formation of colony units is used to assess the frequency of undifferentiated progenitor cells in the whole amount after isolation or at a given stage of culture. The comparison of CFU in cultures allows a comparison between sources. In this way, Angelopoulos et al. [98] established the superior formation of colony units in gingival G-MSCs compared to DP-MSCs. Concerning BM and AT-MSCs, Dmitrieva et al. [88] determined how the CFU ability of AT-MSCs remained stable along passages, but not in BM-MSCs. Thus, it has been shown in another article how the CFU ability of BM-MSCs was 100 times less than other types of MSCs, among other AT-MSCs [89]. Therefore, within cells related to DT, G-MSCs are better than DP-MSCs in attaining better cell efficiency, but between BM and AT-MSCs, AT is the more favorable option. In this sense, both AT and G-MSCs are the superior options once more source productivity is needed.

### 3.4. Angiogenic Capacity

The angiogenic capacity of MSCs has been studied by Angelopoulos et al. [98] in G-MSCs and DP-MSCs. The authors determined that gingival MSCs have more angiogenic capacity than cells from DP. Along this line, vascular endothelial growth factor (VEGF) is a major player in angiogenesis. A general increase in VEGF expression has been observed along with trilineage differentiation of subcutaneous AT-MSCs, omental AT-MSCs, and BM-MSCs [90]. Some authors have shown how important gingival blood perfusion is, along with gingiva regeneration and treatment. Indeed, Alssum et al. [91], in an in vivo study, related the importance of gingival perfusion in gingival and dental regeneration and healing. They showed how vascular and microcirculatory blood flow supply through VEGF, among others, is primordial in oral and dental wound healing. In dental pulp regeneration, vascularization and revascularization are key. In an animal model, Eramo et al. [103] determined how pulp revascularization allowed the regeneration of intracanal pulp-like tissues, with neovascularization, innervation, and dentine formation. VEGF has also been studied in dental pulp regeneration, having an important role in dental pulp and endodontic regenerative medicine [104]. Human VEGF has positive influences on the proliferation, differentiation, mineralization, neovascularization, and formation of reparative dentin of dental pulp tissue [105]. As the oral cavity is composed of both soft and hard tissues, bone regeneration should be considered in oral therapy. Thus, angiogenesis in bone healing and regeneration can be improved by means of MSCs application, as they have been shown to have an angiogenic capacity [106]. Additionally, angiogenic signals are essential to tissue growth and homeostasis, during several regenerative and healing procedures, among others, in periodontal regeneration [18]. In conclusion, in consideration of the importance of cells’ angiogenic capacity to practically all oral regeneration procedures and therapies, G-MSCs, AT-MSCs, and BM-MSCs can be considered indifferently when this target property is desired.

### 3.5. Differentiation Potential

Trilineage differentiation of MSCs is commonly used as a tool for the characterization of MSCs. Bernardo et al. [93], in an assay with MSCs derived from fetal and adult tissue, demonstrated that both fetal and adult cells can differentiate toward the osteogenic, adipogenic, and chondrogenic lineages. In this paper, cells derived from adult and fetal BM had better chondrogenesis than other sources compared (fetal lung and placenta). However, the capability of fetal and adult BM-MSCs to differentiate into chondrocytes, adipocytes, and osteoblasts decreased along passages. In the same sense, Stanko et al. [94] observed the decrease of osteogenic capacity of AT-MSCs and BM-MSCs along passages; in contrast, DP-MSCs have shown an increase in osteogenic capacity along passages. On the other hand, Zhang et al. [92] confirmed that AT-MSCs and BM-MSCs had more osteogenic capacity than PL-MSCs and DF-MSCs. G-MSCs were the source that took longer to differentiate toward the osteogenic lineage. Alternatively, Waldner et al. [91] showed how osteogenic differentiation of omental adipose tissue MSCs and BM-MSCs have higher expression of osteogenic markers than subcutaneous adipose tissue MSCs. Some authors have investigated the expression of key marker genes, differentially expressed during differentiation stages, such as ALP for the osteoblast lineage. Thus, Riekstina et al. [85] showed that BM-MSCs and AT-MSCs are able to constitutively express alkaline phosphatase (ALP) activity with no necessity of induction of differentiation. The expression of ALP was proved to be higher in palatal adipose tissue (PAT-MSCs) than MSCs derived from lipoaspiration [96]. Calcium deposition is also used to test osteogenic differentiation of MSCs, generally by means of specific staining compounds such as Alizarin Red [107,108]. Thus, it was observed that BM-MSCs extracted from human mandibles showed calcium deposition in the extracellular matrix after 7 days of osteogenic induction and mineralization nodes, and stained red alizarin after 21 days of induction [87]. Moreover, Zajdel et al. [109] detected that Alizarin Red staining and ALP activity was higher at 21 days of AT-MSCs differentiation when compared to Warton’s Jelly human umbilical cord MSCs. In this regard, Waldner et al. [91] tested different sources of MSCs (two from AT and one from BM), proving that in BM-MSCs there were superior calcium depositions than in sources related to AT. Winning et al. [99] observed higher expression of ALP and larger calcium deposits, as well as the early expression of differentiation genes (ALP and COL1A1) in PL-MSCs compared to SHEDs and DP-MSCs. In line with these results, Adolpho et al. [110] found that PL-MSCs induced bone formation in rat calvaria defects, independently of the in vitro osteogenic potential. Considering the three sources and their qualities, it is necessary to understand how differentiation is an important ability for dental and oral regeneration. Thus, Hatayama et al. [101] showed how differentiation to fibroblast and keratinocyte cells is central in gingival regeneration. Furthermore, differentiation is a step necessary for dental pulp pathologies [102]. Regarding bone healing and regeneration, Oryan et al. [106] proved the necessity of osteoblast differentiation in bone therapies. In periodontal regeneration, Han et al. [18] observed the importance of differentiation to osteoblast, cementoblast, and fibroblast cells. As it has been revealed, cell differentiation ability is important for most dental and oral applications. Thus, sources such as BM and AT are better than dental tissue at having a good and qualified differentiation, and between AT and BM, BM must be preferred. Among DT-MSCs, those from DP retain osteogenic capacity longer, although those from PL seem to have a greater ability to differentiate.

### 3.6. Secretome

From the earliest studies, the emphasis was on the differentiation potential of mesenchymal stem cells for their application in regenerative medicine, mainly of bone or cartilage. More recently, attention has turned to the molecules that mesenchymal stem cells secrete. Indeed, this secretome appears to have a regulatory role on immunity because, as previously mentioned, secreted soluble molecules influence the behavior of resident endogenous cells. This led Caplan and Correa [111] to coin the expression “medicinal stem cells”, precisely to indicate that mesenchymal stem cells are an in vivo site-regulated “drugstore” of growth factors, cytokines, and other pro-regenerative, immuno-modulatory, anti-inflammatory molecules.

In a recent study, mesenchymal stem cells were engineered by the bone morphogenetic protein-2 gene to produce exosomes. A synergistic effect of the molecules derived from MSCs and up-regulated BMP2 gene expression was demonstrated [112].

Secretome appears to be able to express antibacterial activity, both directly by killing the bacteria and indirectly by stimulating the immune responses of the surrounding resident cells. Studies have shown that it attenuates the inflammatory response to implanted biomaterials [113]. MSC secretome inhibited biofilm formation and the mature biofilm of *S. aureus*, *Staphylococcus epidermidis* (the two major etiological agents of implantation infections), and *Pseudomonas aeruginosa*. MSC secretome cysteine proteases destabilized MRSA biofilms, enhancing the effect of antibiotics previously tolerated by biofilms [114].

An ever-growing body of evidence underlines the importance of MSC secretome in the outcome of cell therapies. Interestingly, MSCs of different origins produce different secretomes, and different secretomes exhibit different regenerative potentials [115].

## 4. Methods

In this review, we have produced an overview of the principles and cutting-edge progress of MSC-based approaches for regenerative medicine applications in dentistry by comparing and focusing attention on the widely used BM-MSCs, AT-MSCs, and the DSCs.

The search for scientific literature was based on up-to-date databases, including Web of Science, Pubmed, Scopus, and Google Scholar. The keywords used in our search were “mesenchymal stem cells”, “bone marrow”, “adipose tissue”, “dental tissue”, “source”, “regenerative medicine”, “proliferation”, “migration”, “angiogenesis”, “CFU”, and “differentiation”. The only criterion for selecting articles was “studies reported in English, because of language barrier”.

This research tried to retrieve reports of relevant research addressing the properties of MSCs derived from bone marrow, adipose tissue, and dental tissue, focusing on the proliferation, migration and angiogenic capacity, formation of colony units (CFU), and differentiation of the cells isolated from these sources.

The results returned 114 papers and 2 book chapters published up to the year 2022. Of these, 109 articles were selected, summarized, and critically discussed to provide a consistent review. Articles written in a language other than English or duplicates were excluded. Figure 2 illustrates the PRISMA flow diagram for the study.

## 5. Conclusions

In this review, we have compared the characteristics of mesenchymal stem cells (MSCs) obtained from different sources to guide clinicians in the identification of the most favorable sources for each clinical application. By means of the analysis of different publications on bone marrow mesenchymal stem cells (BM-MSCs), adipose tissue mesenchymal stem cells (AT-MSCs), and dental tissue mesenchymal stem cells (DT-MSCs), we compared proliferation, migration, and formation of colony-forming unit capacity (CFU), as well as the ability of trilineage differentiation (especially osteogenic ability).

Considering all the characteristics and the differences between sources, it is important to understand the specific properties of all sources to choose which are the best ones from each clinical application. DT-MSCs, BM-MSCs, and AT-MSCs present as a potential source for oral and dental regeneration, as they have been placed as key cells for many of the qualities required in this clinical field. Likewise, in relation to DT-MSCs, G-MSCs can be positioned as the best source derived from DT-MSCs. Finally, in therapies requiring bone regeneration, it may be advantageous to use BM-MSCs as they have better osteogenic capacity than AT-MSCs and DT-MSCs.

## Figures and Tables

**Figure 1 ijms-23-06356-f001:**
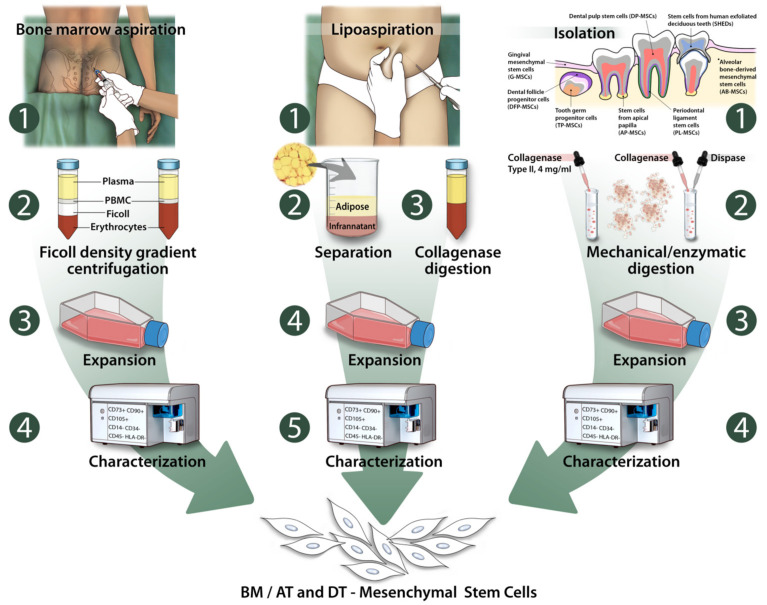
Bone marrow, adipose, and dental tissue as sources of stem cells.

**Figure 2 ijms-23-06356-f002:**
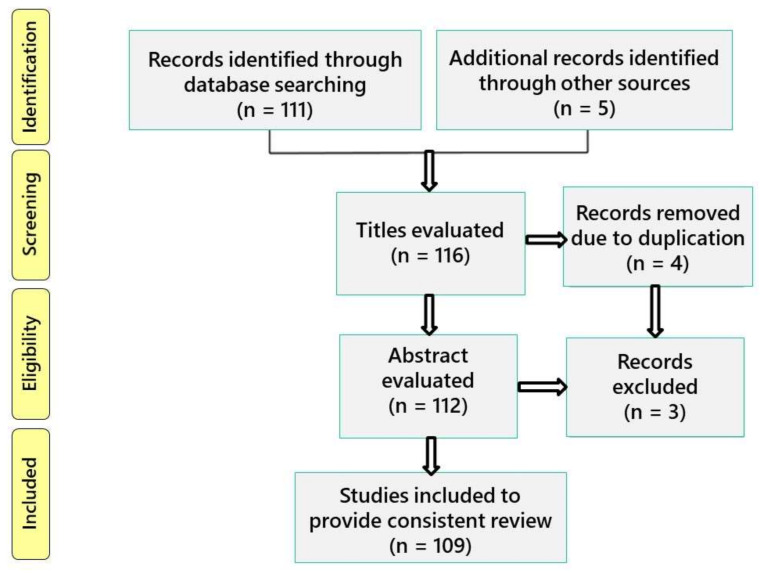
PRISMA flow diagram showing the study selection process.

**Table 2 ijms-23-06356-t002:** Key findings of the MSC Properties in relation to the source.

MSC Source	MSC Property	Key Findings	References
BM-MSCs	Proliferation	BM-MSCs have the potential to double in a 24–72 h period.	Riekstina et al. [85] Heo et al. [86] Zong et al. [87]
	Migration capacity	The migration capacity of BM-MSCs and AT-MSCs is similar.	Jeon et al. [88]
	Clonogenicity	The CFU capacity of BM-MSCs does not remain stable, entering senescence after passage 3–4.	Dmitrieva et al. [89]
		The CFU of BM-MSCs is lower than that of AT-MSCs.	Hayashi et al. [90]
	Angiogenic capacity	VEGF expression has increased alongside differentiation of BM-MSCs	Waldner et al. [91]
	Differentiation potential	BM-MSCs are able to constitutively express alkaline phosphatase (ALP) activity with no necessity of induction of differentiation.	Riekstina et al. [85]
		BM-MSCs extracted from human mandibles showed calcium deposition in the extracellular matrix after 7 days of osteogenic induction and mineralization nodes after 21 days of induction.	Zong et al. [87]
		BM-MSCs showed higher osteogenic capacity compared to PL-MSCs and DF-MSCs.	Zhang et al. [92]
		BM-MSCs carried a higher expression of osteogenic markers than subcutaneous AT-MSCs. Similarly, there were higher calcium depositions in BM-MSCs than in AT-MSC-related sources.	Waldner et al. [91]
		The ability of BM-MSCs (fetal and adult) to differentiate into chondrocytes, adipocytes, and osteoblasts was found to decrease over the passages.	Bernardo et al. [93]
		The osteogenic differentiation capacity of BM-MSCs decreased along the passages.	Stanko et al. [94]
AT-MSCs	Proliferation	AT-MSCs have the potential to double in a 24–48 h period.	Riekstina et al. [85] Heo et al. [86]
		The proliferation rate of human AT-MSCs increases up to passage 10, finding a 32% reduction in proliferation at passage 30.	Danisovic et al. [95]
		There is no significant difference between the proliferation rate of BM-MSCs obtained from lipoaspirate compared to PAT-MSCs.	Hakki et al. [96]
	Clonogenicity	The CFU capacity of AT-MSCs remains stable along the passages.	Dmitrieva et al. [89]
	Angiogenic capacity	VEGF expression has increased alongside differentiation of AT-MSCs (subcutaneous and omental).	Waldner et al. [91]
	Differentiation potential	AT-MSCs are able to constitutively express alkaline phosphatase (ALP) activity with no necessity of induction of differentiation.	Riekstina et al. [85]
		ALP expression was revealed to be higher in PAT-MSCs than in MSCs derived from lipoaspiration.	Hakki et al. [96]
		AT-MSCs showed higher osteogenic capacity than PL-MSCs and DF-MSCs.	Zhang et al. [92]
		AT-MSCs from omental tissue had higher expression of osteogenic markers than subcutaneous AT-MSCs.	Waldner et al. [91]
		The osteogenic differentiation capacity of AT-MSCs decreased along the passages.	Stanko et al. [94]
DT-MSCs	Proliferation	TP-MSCs can be expanded and maintained for almost 60 doubling populations with a high rate of proliferation.	Pandula et al. [69]
		DT-MSCs proliferated faster than BM-MSCs and AT-MSCs	Zhang et al. [92]
		DT-MSCs have a high cell proliferative potential.	Stanko et al. [97]
		G-MSCs showed higher proliferation rates compared to DP-MSCs.	Angelopoulos et al. [98]
	Migration capacity	The migration capacity of G-MSCs is higher than that of DP-MSCs.	Angelopoulos et al. [98]
	Clonogenicity	CFU was higher in G-MSCs compared to DP-MSCs.	Angelopoulos et al. [98]
	Angiogenic capacity	MSCs obtained from gingival tissue showed higher angiogenic capacity than cells from DP.	Angelopoulos et al. [98]
	Differentiation potential	The osteogenic differentiation capacity of DP-MSCs increased along the passages.	Stanko et al. [94]
		PL-MSCs showed increased expression of ALP, calcium deposits, and an early expression of differentiation genes (ALP and COL1A1) compared to SHEDs and DP-MSCs.	Winning et al. [99]

BM-MSCs: Bone marrow stromal cells; AT-MSCs: Adipose tissue derived stem cells; DT-MSCs: Dental tissue-derived stem cells; CFU: colony-forming unit; PAT-MSCs: Palatal adipose tissue stem cells; VEGF: Vascular endothelial growth factor; TP-MSCs: Tooth germ progenitor cells; G-MSCs: Gingival mesenchymal stem cells; DP-MSCs: Dental pulp stem cells; SHEDs: stem cells from the pulp of exfoliated deciduous teeth; PL-MSCs: Periodontal ligament stem cells.

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
