# Peer review of "Different Sources of Mesenchymal Stem Cells for Tissue Regeneration: A Guide to Identifying the Most Favorable One in Orthopedics and Dentistry Applications"

_ijms, 2022, doi:10.3390/ijms23116356_

Round 1
Reviewer 1 Report
The work presented presents the subject matter in an accessible manner. The material presented is correct.
For cells isolated from dental pulp, the proposed enzyme mixture or collagenase II at a concentration of 4 mg/ml can be used. Please indicate both options in the figure.
Author Response
Dear Reviewer,
Thank you for your contribution. Please find below the response to your comments.
Sincerely,
Rebeca Illescas-Montes
Carla Renata Arciola
Reviewer Comments
The work presented presents the subject matter in an accessible manner. The material presented is correct.
- For cells isolated from dental pulp, the proposed enzyme mixture or collagenase II at a concentration of 4 mg/ml can be used. Please indicate both options in the figure.
Response: Thank you for your comments which have helped to improve the article. We have included collagenase II extraction in Figure 1.
Reviewer 2 Report
The authors reviewed the characteristics of Mesenchymal stem cells (MSCs) from different sources with the purpose to help and guide the clinicians to identify the most favorable sources for each clinical application. The authors organized the manuscript in a manner to prove their outcome. However, the manuscript has some unclear issues which demand further clarification.
Please find below some comments/suggestions which might help to improve the quality of the manuscript.
1. The authors inserted as title “Comparison of the Properties of Different Sources of MesenChymal Stem Cells for Regeneration of Bone and Oral Tissues”; however, this title does not fully represent the content of the paper. More particularly, when reading the title and the content of the manuscript are not matching. The authors are requested to reformulate or to double check again the title to be in line with the content.
2. As has been a review a summary of what will be discussed within the manuscript will be welcomed.
3. The authors stated that ” The present review aims to compare the properties of MSCs derived from…”; however, what properties the authors meant to was not mentioned. Moreover, the abstract part was described very generally thus the purpose of the paper was not clear enough; the authors are requested to reformulate the abstract part so that to present briefly aim and scope, methods used and expected outcome.
4. The authors presented MSCs from many sources; however, the compared properties and characteristics were not clear enough. The authors are advised to reformulate so that to be clear the content of the paper. A table summarizing properties vs. sources would have been welcomed.
5. The authors wrote section 4 as methods; however, was not clearly understood what the purpose of this section was. The authors are requested to provide clarifications.
6. Recent literature dating 2022 and 2022 was poor; the authors are requested to provide details of the reason why are not listed recent citations or to update the literature list.
Author Response
Dear Reviewer,
Thank you for your contribution. Please find below the response to your comments.
Sincerely,
Rebeca Illescas-Montes
Carla Renata Arciola
Reviewer Comments
The authors reviewed the characteristics of Mesenchymal stem cells (MSCs) from different sources with the purpose to help and guide the clinicians to identify the most favorable sources for each clinical application. The authors organized the manuscript in a manner to prove their outcome. However, the manuscript has some unclear issues which demand further clarification.
Please find below some comments/suggestions which might help to improve the quality of the manuscript.
Response: Thank you for your insights which have helped to clarify the text and improve the scientific quality of the article. Please find below the responses to your comments:
Comment 1: The authors inserted as title “Comparison of the Properties of Different Sources of MesenChymal Stem Cells for Regeneration of Bone and Oral Tissues”; however, this title does not fully represent the content of the paper. More particularly, when reading the title and the content of the manuscript are not matching. The authors are requested to reformulate or to double check again the title to be in line with the content.
Response: In agreement with the reviewer's comments, we have modified the title. We would expect it to be more in accordance with the content of the review.
Change: We have changed the title to: “Different sources of mesenchymal stem cells for tissue regen-eration: A guide to identifying the most favorable one in or-thopedics and dentistry applications”
Comment 2: As has been a review a summary of what will be discussed within the manuscript will be welcomed.
Response: Thank you very much for your comment. We have modified the summary to include the important aspects discussed in the article.
Comment 3: The authors stated that ” The present review aims to compare the properties of MSCs derived from…”; however, what properties the authors meant to was not mentioned. Moreover, the abstract part was described very generally thus the purpose of the paper was not clear enough; the authors are requested to reformulate the abstract part so that to present briefly aim and scope, methods used and expected outcome.
Response: In consonance with the previous revision. We have modified the abstract to clarify the properties of the MSC properties targeting of study, as well as the methodology and main results:
Change: “…The present review aims to compare of MSCs properties (proliferation, migration, clonogenicity, angiogenic capacity, differentiation potential, and secretome) derived from bone marrow, adi-pose tissue, and dental tissue to enable clinicians to select the best source of MSCs for their clinical application in bone and oral tissue regeneration to delineate new translational perspectives. A re-view of the literature was conducted using the search engines Web of Science, Pubmed, Scopus, and Google Scholar. By means of the analysis of different publications, it shows that all sources compared [bone marrow mesenchymal stem cells (BM-MSCs), adipose tissue mesenchymal stem cells (AT-MSCs), and dental tissue mesenchymal stem cells (DT-MSCs)] are good options to pro-mote proper migration and angiogenesis, and they turn out to be useful for gingival, dental pulp, bone, and periodontal regeneration. In particular, DT-MSCs have better proliferation rates and AT and G-MSC sources showed higher clonogenicity. MSCs from bone marrow, widely used in orthopedic regenerative medicine are preferable for their differentiation ability. Considering all the properties among sources, BM-MSCs, AT-MSCs, and DT-MSCs present as a potential candi-dates for oral and dental regeneration”
Comment 4: The authors presented MSCs from many sources; however, the compared properties and characteristics were not clear enough. The authors are advised to reformulate so that to be clear the content of the paper. A table summarizing properties vs. sources would have been welcomed.
Response: In accordance with the recommendation, Table 2 has been added in order to clarify the comparison of the properties of the different sources.
Comment 5: The authors wrote section 4 as methods; however, was not clearly understood what the purpose of this section was. The authors are requested to provide clarifications.
Response: Thank you for your review. The methods section aimed to describe the process of searching for scientific literature, as well as the selection procedure of the papers. We have rewritten parts of this section to make it clearer.
Comment 6: Recent literature dating 2022 and 2022 was poor; the authors are requested to provide details of the reason why are not listed recent citations or to update the literature list.
Response: Thank you for your review. We have updated the literature by adding recently published papers:
- Ding, G.; Du, J.; Hu, X.; Ao, Y. Mesenchymal Stem Cells From Different Sources in Meniscus Repair and Regeneration. Front Bioeng Biotechnol 2022, 10, 796367, doi:10.3389/fbioe.2022.796367.
51. Salamanna, F.; Contartese, D.; Borsari, V.; Pagani, S.; Barbanti Brodano, G.; Griffoni, C.; Ricci, A.; Gasbarrini, A.; Fini, M. Two Hits for Bone Regeneration in Aged Patients: Vertebral Bone Marrow Clot as a Biological Scaffold and Powerful Source of Mesenchymal Stem Cells. Front Bioeng Biotechnol 2021, 9, 807679, doi:10.3389/fbioe.2021.807679.
- Zong, C.; Zhao, L.; Huang, C.; Chen, Y.; Tian, L. Isolation and Culture of Bone Marrow Mesenchymal Stem Cells from the Human Mandible. JoVE (Journal of Visualized Experiments) 2022, e63811, doi:10.3791/63811.10
110. Adolpho, L.F.; Lopes, H.B.; Freitas, G.P.; Weffort, D.; Campos Totoli, G.G.; Loyola Barbosa, A.C.; Freire Assis, R.I.; Silverio Ruiz, K.G.; Andia, D.C.; Rosa, A.L.; et al. Human Periodontal Ligament Stem Cells with Distinct Osteogenic Potential Induce Bone Formation in Rat Calvaria Defects. Regen Med 2022, 17, 341–353, doi:10.2217/rme-2021-0178.

Round 2
Reviewer 2 Report
The authors answered to the addressed queries and updated the manuscript accordingly. Thank you.